An approach to fill in missing data from satellite imagery using data-intensive computing and DINEOF

Lomelí-Huerta José Roberto 1
http://orcid.org/0000-0003-3188-1448 Rivera-Caicedo Juan Pablo 2
http://orcid.org/0000-0002-0937-5656 De-la-Torre Miguel 1
Acevedo-Juárez Brenda 3
Cepeda-Morales Jushiro 4
http://orcid.org/0000-0001-8578-0170 Avila-George Himer 1 himer.avila@academicos.udg.mx
1 Departamento de Ciencias Computacionales e Ingenierías, Universidad de Guadalajara , Ameca, Jalisco , México
2 CONACYT-UAN, Secretaría de Investigación Posgrado, Universidad Autónoma de Nayarit , Tepic, Nayarit , Mexico
3 Departamento de Ciencias Naturales y Exactas, Universidad de Guadalajara , Ameca, Jalisco , Mexico
4 Centro Nayarita de Innovación y Transferencia de Tecnología A. C., Universidad Autónoma de Nayarit , Tepic, Nayarit , Mexico
Mei Gang
Electronic publication date: 2022 May 13
Publication date: 2022
Volume: 8
Electronic Location ID: e979
Received 2020 Nov 11; Accepted 2022 Apr 22
Copyright: © 2022 Lomelí-Huerta et al.
Copyright year: 2022
Copyright holder: Lomelí-Huerta et al.
License: This is an open access article distributed under the terms of the Creative Commons Attribution License, which permits unrestricted use, distribution, reproduction and adaptation in any medium and for any purpose provided that it is properly attributed. For attribution, the original author(s), title, publication source (PeerJ Computer Science) and either DOI or URL of the article must be cited.
License URL: https://creativecommons.org/licenses/by/4.0/

Keywords: Satellite imagery, Missing data, DINEOF, MODIS, VIIRS

Funding: CONACYT-INEGI 290628 This research was financed by the project: CONACYT-INEGI 290628. The funders had no role in study design, data collection and analysis, decision to publish, or preparation of the manuscript.

==============================
This paper proposes an approach to fill in missing data from satellite images using data-intensive computing platforms. The proposed approach merges satellite imagery from diverse sources to reduce the impact of the holes in images that result from acquisition conditions: occlusion, the satellite trajectory, sunlight, among others. The amount of computation effort derived from the use of large high-resolution images is addressed by data-intensive computing techniques that assume an underlying cluster architecture. As a start, satellite data from the region of study are automatically downloaded; then, data from different sensors are corrected and merged to obtain an orthomosaic; finally, the orthomosaic is split into user-defined segments to fill in missing data, and filled segments are assembled to produce an orthomosaic with a reduced amount of missing data. As a proof of concept, the proposed data-intensive approach was implemented to study the concentration of chlorophyll at the Mexican oceans by merging data from MODIS-TERRA, MODIS-AQUA, VIIRS-SNPP, and VIIRS-JPSS-1 sensors. The results revealed that the proposed approach produces results that are similar to state-of-the-art approaches to estimate chlorophyll concentration but avoid memory overflow with large images. Visual and statistical comparison of the resulting images revealed that the proposed approach provides a more accurate estimation of chlorophyll concentration when compared to the mean of pixels method alone.

Introduction

Since the first satellite photographs in the 1940s, followed by missions that include Landsat and Suomi NPP from NASA, and Sentinel from ESA, satellite imagery has been improved to the point of becoming daily-use information. Moreover, together with the increase of use, challenges have been emerged, presenting an increasing demand for computational resources and algorithms. Nowadays, images of the Earth are commonly used to study the atmosphere, land, and oceans, presenting an increasing use in daily life activities. Applications of satellite imagery range from weather forecasting (Sato et al., 2021), monitoring natural disasters (Said et al., 2019), survey phytoplankton size structure impacts as an ecological indicator for the state of marine ecosystems (Gittings et al., 2019), among many others. Presently, various sensors provide different temporal, spatial, and spectral resolutions to study oceans’ evolution. The Moderate-Resolution Imaging Spectroradiometer (MODIS) and Visible Infrared Imaging Radiometer Suite (VIIRS) sensors are well known in the community, mainly because of data’s continuous and free availability. Data from these sensors have been available since 1999 and are still in operation (Hu et al., 2010). The MODIS sensors are in orbit aboard the Terra and Aqua satellites (NASA, 2020). On the other hand, the VIIRS sensors are aboard the Suomi National Polar-Orbiting Partnership (SNPP) and the Joint Polar Satellite System (JPSS-1) (Kramer, 2020). Both MODIS and VIIRS sensors are provided with a set of bands commonly employed to study the oceans (Datla et al., 2016). Regardless of the application, processing data from satellites exhibit various challenges that are difficult for the analysis related to physical phenomena (Rodriguez-Ramirez et al., 2019). Moreover, one of the most representative issues emerge from acquisition conditions that produce incomplete data from the whole scene, either caused by occlusion (e.g. clouds) or the trajectory of the satellite at the acquisition moment (Zhang et al., 2018).

As reported by the scientific community, some approaches to fill in missing data employ machine learning techniques to merge information from multiple sources within the same set of sensors (Zhang et al., 2018). Other approaches use regression models to repair single spatial satellite images, presenting a tradeoff between accuracy and computational effort (Zhang, Clayton & Townsend, 2015). Furthermore, one of the most widely employed methods in oceanography to fill in missing data is DINEOF (Data INterpolating Empirical Orthogonal Function) (Liu & Wang, 2018; Liu & Wang, 2019). Examples of the use of DINEOF to study chlorophyll are disseminated in literature. For instance, Jayaram et al. (2018) implemented the interpolation functions of the orthogonal data to restore the levels of chlorophyll-a (Chl-a) at the Arabic sea between 2000 and 2015, using the MODIS sensor. Jayaram et al. (2021) compute the chlorophyll concentration using DINEOF to fill in gaps produced by clouds, using the Ocean Colour Monitor-2 (OCM-2) onboard Oceansat-2 satellite for the period 2016–2019 over the northern Indian Ocean. On the other hand, Alvera-Azcárate et al. (2011) implemented the data restoration with DINEOF using the time series with a single variable (monovariate), and several variables (multivariate approach). More recently, Alvera-Azcárate et al. (2021) reported a suspended particulate matter reconstruction combining Sentinel-2 and Sentinel-3 imagery using DINEOF. The advantage of such a combination allows us to retain both the high spatial resolution of the Sentinel-2 data while increasing the temporal resolution from Sentinel-3 data. DINEOF was also employed by Bouchra et al. (2011) to restore the total suspended matter between Belgium and United Kingdom coasts, using MODIS data acquired between 2003 and 2006. Restored data were compared against the measurements in-situ of total suspended matter collected by the Centre for Environment Fisheries and Aquatic Sciences (Cefas); for factor calibration, a linear regression model was employed, considering the highest observed measurements as the reference values. Additionally, during the atmospheric correction, MODIS data pixels were labeled according to the quality of the restoration: those pixels within a 5 × 5 window that present inconsistencies over the Cefas time series were labeled as doubedly or low quality. Finally, DINEOF was used to compute missing data, and atypical values were assessed using spatial coherence.

Despite the approach employed to fill in missing data, and the study region, the challenges remain. Furthermore, improvements in computational efficiency and accuracy are still required to produce reliable studies. Indeed, the high computational cost required to analyze multi-temporal and multi-resolution data provided by satellite platforms is far from being solved (Babbar & Rathee, 2019). In particular, DINEOF is based on empirical orthogonal functions (EOF) to reconstruct missing data in a set of geophysical data through the calculus of the dominant modes of variability within satellite data (Beckers & Rixen, 2003). The DINEOF’s amount of computation increases with the size of the input images and may be impractical with a large number of high-resolution images. Therefore, DINEOF is usually used to process images with a low spatial resolution of small geographical areas (GHER, 2020).

This research article proposes a novel approach for automated hole filling in satellite imagery. The first novelty of the proposed approach compared to previous works is that it uses data from different sensors, while previous works used data from the same set of sensors. Another novelty in our proposal is the way the filling of missing data was performed, which is carried out by chaining three different strategies: (1) The first data with which the gaps in the images are filled comes from the fusion of four data sources (MODIS-TERRA, MODIS-AQUA, VIIRS-SNPP, and VIIRS-JPSS-1); (2) the next step consists of estimating the missing pixels close to those obtained in the previous step, for which the nearest neighbor approach using multivariate interpolation is employed; and (3) empirical orthogonal functions are used to fill in the last missing data. Finally, the proposed approach uses an intensive computing strategy to avoid memory overflow when processing high-resolution images. For proof of concept, the detection of chlorophyll over the exclusive economic zone of Mexico (EEZM) is analyzed.

A computer-intensive approach to fill in missing data

The proposed data-intensive computing approach to fill in the missing geophysical data from satellite imagery comprises three main conceptual modules: (1) automatic satellite data download, (2) satellite data merging, and (3) filling in missing satellite data using an intensive computer approach. Each module considers the output of the previous one, and their operation is detailed in the sections below. The whole process is depicted in Fig. 1.

Figure 1 Proposed approach for filling in missing satellite data.

First, L2 data from the region of interest (ROI) were downloaded. Satellite data were then merged using a 2-step strategy, estimating missing pixels as the average of at least three neighbors in same-day images from different sensors. Finally, missing data was filled in using a data-intensive approach that takes advantage of segmented ROIs and DINEOF.

Automatic satellite data download

The automatic satellite data download module is designed to continuously survey changes in the satellite repository and retrieve the most recent satellite imagery from the region of interest (ROI). Without loss of generality, it is assumed that data are retrieved from the OBPG-Ocean color data repository, but other platforms may be configured with the same behavior. The three steps established in this module are listed below. The first step consists in querying the repository to retrieve the schedule of both sensors (MODIS and VIIRS): the time when they passed over the zone of study.

In the second step, the schedule information is processed to extract the precise hour when the satellite acquired the region of interest (ROI).

Finally, the links to the levels L2 products are built in the third step, and the download process starts. The resulting L2 products are stored in a user-specific path that is accessed by the other two modules.

As a result, the module for satellite data download retrieves the high-resolution images from the configured sensors, corresponding to the ROI at a determined date.

Satellite data merging

Every time L2 satellite data corresponding to the ROI are downloaded, a new daily high-resolution image is created by merging data from the selected sensors according to the application. The procedure to create the combined image involves the two steps described in subsections below (preprocessing satellite data and merging preprocessed data).

Preprocessing satellite data

The first novelty of the proposed approach is that data from different sensors are used when performing satellite data fusion. Preprocessing data consists of creating the orthomosaics for each daily scene: one for each sensor (I1, I2, ⋯ IL). Operations like spatial resampling or scaling are required in some cases to prepare the raw data to assemble a single orthomosaic for each of the L sensors. Subsequently, each orthomosaic is processed to fill missing data during the merging phase; a m × m sliding window is applied to the p empty pixels in the image that accomplish with the criteria of having at least three neighbors (i.e., three pixels with data). Such a criterion was established to avoid simple information duplicity of close pixels. The new value px of an empty pixel is computed using Eq. (1).

(1) px=∑i=1npin,

where px is the missing data pixel, pi is one of the n neighbor pixels with data within the m × m sliding window, considering n ≥ 3.

Merging preprocessed data

In this step, the preprocessed orthomosaics are merged. In order to obtain the combined image at a selected date (day), the orthomosaic with most data related to chlorophyll is first chosen and tagged as base-image (Ib). Then, it is necessary to define the order of the processing of each orthomosaic. The ordering criteria considers the root-mean-square error (RMSE) between the base-image and each of the remaining orthomosaics (Ir), assigning higher priority to the orthomosaics with lower RMSE, see Eq. (2).

(2) RMSE(Ir)=1N∑r=13(Ib−Ir)2

where Ib is the base-image, Ir corresponds to each of the other images, and N is the number of valid pixels in both images (i.e., Ib and Ir).

Once the priority is established, data from the four images are combined considering Ib as the baseline and following the order of priority given by the RMSE: each missing pixel with coordinates (x, y) in Ib is substituted with the pixel from Ir with the highest priority on the same position. If none of the Ir images contains data, it is considered as a missing pixel. Finally, an adjustment is applied to reduce the impact of differences in acquisition conditions from each sensor, such as different acquisition times and the zone dynamics (currents and winds). Such an adjustment between images Ib and Ir was applied using the Inverse Distance Weighting (IDW) to the four nearest pixels in directions (−x, x, −y, and y). In essence, the resulting high-resolution image IM produced by merging the sensor-wise orthomosaics {I1, I2, I3, I4} incorporate the information from all sensors, and hence, includes fewer gaps than any of the individual orthomosaics.

Filling in missing satellite data

As a second contribution, the proposed approach is able to process high-resolution images of large study areas. After preprocessing, the merged orthomosaic IM still remains with gaps, and DINEOF is employed to compute and fill in the gaps. In order to address this problem with high-resolution images from wide areas of study, the data-intensive approach is divided into the following three steps: (1) data segmentation, (2) fill in missing data for each segment, and (3) assemble the segments. The strategy to fill in missing data is shown in Fig. 2, and each step is detailed in sections below.

Figure 2 Filling in missing satellite data takes advantage of parallel processing to independently process previously divided segments and assemble results in a single orthomosaic.

Data segmentation

The merged orthomosaic IM comprises the whole ROI to be monitored, which may be computationally unmanageable, depending on the area of study and the computer to process DINEOF. Thus, IM is evenly divided into J × K = NS smaller manageable size segments {ISi,j}:

(3) IM=[I1,1SI1,2S⋯I1,KSI2,1SI2,2S⋯I2,KS⋮⋮⋱⋮IJ,1SIJ,2S⋯IJ,KS]

The user-defined values for J and K should be selected according to the computational resources available to execute DINEOF, and indirectly define the size of the segments {ISj,k}. Inspired by binary search, the orthomosaic may be evenly divided into 2 × 2, 4 × 4, 8 × 8, and so on. As soon as the computer system is able to process the images, the divisions are fixed and the monitoring process configured.

Fill in missing data for each segment

Filling in missing data is a parallel process that is independently applied to all segments in which the merged image was divided (see Fig. 2). Using a massive processing configuration (e.g. a computer cluster or a multiprocessor computer) is advantageous to accelerate the complete process. In this step, each segment ISj,k is filled in, and the resulting filled segments IFj,k are stored for posterior processing.

Assemble segments

At the final module, the resulting IFj,k segments are assembled in the same order that was divided IM, to obtain a new IF orthomosaic without holes.

(4) IF=[I1,1FI1,2F⋯I1,KFI2,1FI2,2F⋯I2,KF⋮⋮⋱⋮IJ,1FIJ,2F⋯IJ,KF]

A distinct but equivalent way to define the number of segments is to establish the size of each segment IFj,k, assuming all segments are the same size. The size of IFj,k corresponds to a 2-element tuple (width, height) that define the number of pixels per side, considering the ratio between width and height of the segment to be the same of the ratio between the width and the height of the orthomosaic IM: width(Ij,kF)height(Ij,kF)=width(IM)height(IM).

Computational complexity analysis

The application of DINEOF to a sequence of T orthomosaics requires to assembly a L × T matrix, with L = width × height representing the number of pixels in IM. After that, the resulting matrix is standardized, and the optimal number of empirical orthogonal functions (EOFs) are by the convergence of a validation process that depends on Singular Value Decomposition (SVD) computation. The computation of SVD is in the order O(LT2), and the validation process depends on the maximum number of iterations (Q) employed to find the optimal number of EOFs. Thus, the whole computation of DINEOF for a sequence of T orthomosaics is in the order O(QLT2), with typical values of L ≪ T and L ≫ Q: the number of pixels usually greatly exceeds the time frame T, as well as the iterations Q. Consequently, a significant reduction in the number of pixels L per segment ISj,k causes a consequent reduction in the total number of operations.

Study case: chlorophyll on the eezm

The study case used for proof of concept was designed to monitor the Chl-a over a wide sea area: EEZM. Data from the MODIS and VIIRS sensors were combined to obtain L2 products with the least amount of missing data. The importance of monitoring the Chl-a is related to the dynamics of phytoplankton, which provides the information to predict the impact of climate change in ocean ecosystems. Phytoplankton is composed of microscopic algae and other photosynthetic organisms that inhabit the surface of oceans, rivers, and lakes. These microorganisms constitute the primary source of energy in aquatic systems due to their photosynthetic capacity (Winder & Sommer, 2012), and their contribution to preserving the climate balance and the biogeochemical cycle in such ecosystems (Hallegraeff, 2010). For some decades now, the Chl-a has been widely used to estimate phytoplankton’s biomass in surface water using satellite-based methods (Gomes et al., 2020; Kramer & Siegel, 2019; O’Reilly et al., 1998). Such usage is given in view of the fact that the Chl-a is the main photosynthetic pigment of phytoplankton. In fact, Chl-a is used as a photoreceptor and gives the green color to the phytoplankton, and various studies have settled the fundamentals of the impact of Chl-a with light reflectance of water bodies, especially in the visible light and close infrared regions of the electromagnetic spectrum (Gitelson, 1992; Dall’Olmo & Gitelson, 2005; Yacobi et al., 2011).

Study area

The area of study selected for the analysis and proof of concept corresponds to the EEZM, and covers the sea region close to the seashore (CONABIO, 2022). The distance covered by the EEZM is up to 370.4 km from the continental and insular seacoast. The surface area of the EEZM is one of the greatest in the world and is estimated to be 3,269,386 km2.

The complete satellite images are required to study Chl-a concentrations, e.g. images without missing data over the area of study. The size of such a huge area makes the task prohibitive for the computer facilities available for experimentation. Bands 8 to 16 from the MODIS sensors were used, corresponding to wavelengths from 405 to 877 nm and a spatial resolution of 1 km. These bands are mainly employed for ocean color and phytoplankton and biogeochemistry. On the other hand, the VIIRS sensor provides measurements from water, land, and atmosphere, with a temporal resolution of 12 h for day and night ocean data acquisition.

Computational details on experiments

A multiprocessor computer with distributed memory was used to run the experiments. The so called Perseo computer, is part of the computing network of the Centro Nayarita de Innovación y Transferencia de Tecnología A.C., México. The Perseo cluster is provided with 388 processing cores, 1,280 GB Ram, 356 TB permanent storage, and runs the CentOS 7.0 operating system. The proposed approach was implemented using a combination of scripts written in Python and Matlab 2018. In particular, the automated image download module written in Python, and the whole processing code written in Matlab are freely available to download through GitHUB: https://github.com/jroberto37/fill_missing_data.git. The download script takes advantage of the geolocation products that include MOD03, MYD03, VNP03MODLL, and VJ103DNB, for sensors MODIS-TERRA, MODIS-AQUA, VIIRS-SNPP, and VIIRS-JPSS-1 respectively.

For preprocessing, the Graph Processing Tool (GPT) from the Sentinel Application Platform (SNAP) was used to create the orthomosaics and project the sine wave system’s data to the WGS-84. Segmentation was performed over the merged high-resolution image to speed up the process of filling chlorophyll concentration data, following the NetCDF format. The maximum area of the segments is defined in the system configuration parameters and automatically establishes the number of segments in which the image is divided. The *.gher binary files are then generated with their respective mask of the zone that is not processed (e.g., land), as well as its time file that allows activating the filtering of the temporal covariance matrix.

The *.gher and time files generated at segmentation are then used to execute DINEOF, employing the configuration parameters shown in Table 1. The proposed algorithm rewrites such parameters in a file with the *.init extension. Afterward, the file is read by the DINEOF program, which computes the missing data for each of the segmented data series. In the fill-in missing data step, DINEOF generates a time series without holes for each segment, and it is stored in *.gher file format. Finally, the orthomosaic is reconstructed using the segmented high-resolution images without missing data. The resulting image is written in NetCDF format.

Table 1 Parameters employed in the evaluation of DINEOF.

Parameter	Description	Range	
alpha	Parameter specifying the strength of the filter	[3 5 10 20 50]	
numit	Number of iterations for the filter	[0.1 0.3 0.5]	
nev	The maximum of number of modes you allow to compute	20	
neini	The minimum number of modes you want to compute	1	
ncv	The maximal size for the Krylov subspace	35	
tol	The threshold for Lanczos convergence	1.0e−8	
nitemax	The maximum number of iteration allowed for the stabilization of eofs obtained by the cycle	300	
toliter	Precision criteria defining the threshold of automatic stopping of DINEOF iterations	1.0e−3	
rec	For complete reconstruction of the matrix	0	
eof	Writing the left and right modes of the input matrix	0	
norm	Activate the normalization of the input matrix	0	
seed	Seed to initialize the random number generator	243435	

Evaluation in cloudy scenarios

For validation, a free of the holes data set was generated for the time frame from January 2017 to December 2019. The data set was composed of 36 complete high-resolution images (without missing pixels), with a spatial resolution of 1 km from the four sensors. The images were composed with the 30 Chl-a daily images from each month. As an example, Fig. 3A shows that the Chl-a composed image corresponding to January 2018 does not present black or white regions, corresponding to zones with missing data.

Figure 3 Maps of cloud masks in Mexico economic exclusive zone (EEZM).

(A) Chl-a image composed by 30 scenes (January 2018), (B) mask with 20% clouds, (C) mask with 30% clouds and (D) mask with 50% clouds.

Three scenarios were prepared to test the system under occlusion conditions by adding different levels of synthetic clouds to the composed images. The three levels of missing data were arbitrarily selected to represent different typical scenarios that are common in real data. Figures 3B–3D show the composed image corrupted with the synthetic cloud masks, covering 20%, 30%, and 50% respectively. The generation of synthetic masks was based on real cloud images from the same scene at different dates (e.g. climate conditions), and the percentage of clouds was computed based on pixel counts. Regarding the cloud coverage in Fig. 3B, a few clouds scarcely cover different regions of the sea, shaping natural clouds. Increasingly dense clouds are shown in Figs. 3C and 3D, according to the corresponding percentage of the cloud masks.

Experimental results

Satellite data merging

The sensibility of the preprocessing satellite data module to the size of the sliding window was studied using nine different square windows: m × m windows with m = {3, 5, 7, 9, 11, 15, 21, 31 and 51}. In this sensibility test, the base image employed for each sensor was composed by the sequence of images for February 2018; and missing samples were generated using the images for February the 1st, 2018, for each sensor.

Table 2 shows the RMSE, the percentage of filled data, the computation time that was employed in the nine different window sizes, and the mosaicking SNAP function. The mosaicking results are the reference for data coverage previous to the application of the sliding window. According to Table 2, the window sizes 5 × 5 and 7 × 7 presented the lowest RMSE value when compared to other window sizes. However, the latter showed a higher percentage of data coverage (28.71% against 25.52%), although the processing time increases according to the window size.

Table 2 Results of the preprocessing module in terms of RMSE, percentage of data coverage in the resulting orthomosaic, and the preprocessing time in seconds.

	Mosaicking	3 × 3	5 × 5	7 × 7	9 × 9	11 × 11	15 × 15	21 × 21	31 × 31	51 × 51	
RMSE	0.429	0.419	0.408	0.408	0.421	0.437	0.471	0.513	0.582	0.674	
Data coverage (%)	17.35	20.98	25.53	28.72	30.57	32.31	35.01	37.96	41.37	45.56	
Time (s)	7.753	8.748	13.402	17.583	19.314	21.640	28.685	40.228	60.736	138.663	
Note:

Bold numbers symbolize the best results when distinct window sizes are compared.

In order to compare the results of the mosaicking and evidence the advantages of applying the sliding window, Fig. 4 shows the results of the application of the method to high-resolution images for February 1st, 2018. The four images in Fig. 4 correspond to the central region of the Gulf of Mexico, with coordinates North = 30.40°, South = 18.60°, East = −83.30° and West = −94.90°. Figure 4A shows the original image from sensor VIIRS-JPSS-1. Figure 4B shows the spatial distribution of the pixels from both methods and the pixels that were filled with the proposed approach. Figure 4C shows the results of the mosaicking function from the SNAP software; and Fig. 4D shows the results obtained with the window size 7 × 7. A visual comparison of Figs. 4D and 4A evidences the advantage of using the sliding window to reduce the amount of missing data, even when compared to commercial software (Fig. 4C). Images from the four sensors were complemented with different percentages of missing data: 67.72% for MODIS-AQUA, 65.93% for MODIS-TERRA, 58.55% for VIIRS-SNPP, and 58.29% for VIIRS-JPSS-1.

Figure 4 Impact of the pre-filling with sliding windows in the region of the Gulf of Mexico.

(A) JPSS-1 original, (B) filling zones, (C) mosaicking and (D) Windows 7 × 7.

Figure 5 shows the orthomosaics obtained for each sensor after preprocessing downloaded samples and applying the 7 × 7 sliding window. Due to the differences in trajectories and climate conditions at the overflight time, all orthomosaics present quite different areas of missing samples. For example, Figs. 5A and 5B corresponding to MODIS Aqua and MODIS Terra respectively, have a band of missing data at the center of the image, but with distinct orientations. On the other hand, Figs. 5C and 5D that correspond to VIIRS JPSS-1 and SNPP, do not present clear missing data patterns. Such differences favor the exploitation of the different sources to obtain a more complete resulting orthomosaic IM. Once the four orthomosaics were generated, data from the VIIRS-JPSS-1 sensor was selected as the base image in the merging preprocessed data module. The orthomosaic from the VIIRS-JPSS-1 sensor was chosen as the base image (Ib) because it presents a lower percentage of missing data than the other sensors. Then, the final merged IM is processed with DINEOF with the orthomosaics from previous days, as described in the following section.

Figure 5 Maps of orthomosaics in Mexico economic exclusive zone (January 1st 2019).

(A) MODIS Aqua orthomosaic, (B) MODIS Terra orthomosaic, (C) VIRSS JPSS-1 orthomosaic and (D) VIRSS SNPP orthomosaic.

Filling in missing satellite data

After preprocessing was applied to the whole temporal series 2018–2019, the impact of the three hyperparameters was evaluated on the proposed system: (1) alpha, (2) numit, and (3) time (see Table 1). The values of the hyperparameters were explored through the application of DINEOF, after splitting the preprocessed orthomosaic into six independent zones (see Fig. 6). Such a division favors the analysis of Chl-a’s behavior either in coastal zones or deep sea, separated from coasts. For example, zone 4 presents deep seawater with a concentration of Chl-a that differs from the concentration in zone 5, which is closer to coasts. On the other hand, a high concentration of Chl-a can be observed close to the coasts in zones 1 to 4 and 6. In that sense, Fig. 6 presents the six zones in which the area of study was divided for the evaluation of the DINEOF tuning parameters.

Figure 6 Segmentation of the area of study, performed automatically by the proposed approach.

The proposed approach was evaluated at two different levels. First, at the adjustment of internal hyperparameters of DINEOF, where image segmentation was adapted to 2 × 3 sub-images, and hyper-parameters alpha and numit were evaluated according to ranges in Table 1. The search for the more suitable hyperparameters for DINEOF was conducted by running two experimental designs: one for time = 30, and another for time = 60. During the adjustment process, the RMSE was estimated for the distinct possible values of alpha and numit, considering the ranges established in Table 1.

The resulting expected error obtained through the search process is shown in Fig. 7. The first and third rows of Fig. 7 (images a, b, c, g, h, and i) represent the expected error for the temporal series with time = 30. Similarly, the second and fourth rows of Fig. 7 (images d, e, f, j, k, and l) represent the expected error for the temporal series with time = 60. In all images from Fig. 7, the horizontal axis represents the alpha parameter, the vertical axis represents the numit parameter, and the color of the cells represents the expected error computed with DINEOF. The color scale is shown at the bottom of the same figure. According to Figs. 7B and 7E, the highest expected error was attained at zone 2, either with time = 30 or time = 60. And in those cases, the values of alpha = 0.1, and numit = 3.0 present a lower expected error, e.g., seems to be favorable in both scenarios. On the other hand, Figs. 7G and 7J exhibit the lowest expected error, regardless of the value assigned to both alpha and numit, as well as the timeframe. In the rest of the cases and regardless of the time frame, the lowest expected error is attained with alpha = 0.1 and numit = 3.0, and they were fixed for the application of the segmented fill in the algorithm.

Figure 7 RMSE corresponding to the application of DINEOF for different values of alpha and numit, at distinct zones and time frames; the colorbar represents the RMSE.

(A) Zone 1/time = 30, (B) Zone 2/time = 30, (C) Zone 3/time = 30, (D) Zone 1/time = 60, (E) Zone 2/time = 60, (F) Zone 3/time = 60, (G) Zone 4/time = 30, (H) Zone 5/time = 30, (I) Zone 6/time = 30, (J) Zone 4/time = 60, (K) Zone 5/time = 60 and (L) Zone 6/time = 60.

With fixed hyperparameters, at the second level of evaluation, the RMSE was computed for distinct areas of the segments on the three cloudy scenarios (e.g. 20%, 30%, and 50% clouds). Four segmentation levels were considered for IM in order to parallelize the process, with image segments represented by the triplets j × k × t, with j and k as described in Section Filling in missing satellite data; and t representing the time in trimesters. The segmentation levels correspond to 312 × 187 × 12, 625 × 374 × 12, 1,250 × 749 × 12, and 2,500 × 1,498 × 12. However, the computer configuration employed to run the software was not able to completely run the system with the latter configuration due to memory overflow.

Table 3 presents the average time and RMSE that were obtained after the execution of the experimentation with the aforementioned segment sizes. According to Table 3, the computation of the missing data showed a better performance when the segments size and the amount of data to estimate were rather small compared to the size of IM. In fact, the lowest RMSE was attained when IM was divided into 8 × 8 segments in the three cloud scenarios. In the hardest scenario, with 50% of missing data due to clouds, the proposed approach achieved an RMSE of 0.43, which was lower than all other feasible cases.

Table 3 Average time and RMSE obtained after the application of the proposed approach with distinct segment sizes.

Segment size	312 × 187 × 12	625 × 374 × 12	1,250 × 749 × 12	
Cloud test (%)	Time (σ)	RMSE (σ)	Time (σ)	RMSE (σ)	Time (σ)	RMSE (σ)	
20	7.52 (4.59)	0.45 (0.11)	43.41 (28.82)	0.36 (0.08)	187.17 (111.09)	0.37 (0.08)	
30	10.42 (27.84)	0.47 (0.12)	43.21 (27.76)	0.35 (0.07)	175.84 (90.69)	0.36 (0.06)	
50	13.86 (31.77)	0.52 (0.16)	58.36 (42.02)	0.43 (0.17)	186.72 (130.17)	0.44 (0.16)	
Note:

Bold numbers symbolize the lowest RMSE.

Finally, Fig. 8A presents the merged image IM, created with the data acquired by the MODIS and VIIRs sensors on January 1st, 2019. On the other hand, Fig. 8B presents the final result of the proposed approach, which was created with satellite data from June 1st, 2018 to May 5th, 2019. It can be observed that there are no holes or missing data, and it is ready to create time-series related to the concentration of Chl-a.

Figure 8 Results of the process of filling in missing data in satellite images.

(A) Merged image (IM). (B) Final image without missing data (IF).

Computational complexity analysis

As an example that follows the case study, processing the sequences with the original size (2,500 × 1,498 × 12) requires as many operations as O(300 × 3,745,000 × 122). On the other hand, by applying the proposed segmentation strategy, the task is divided into 16 sequences of 312 × 187 × 12, which require as many operations as O(300 × 58,344 × 122). Following this example, Fig. 9 presents the number of operations expected for each segment size, showing the effect of the split of the whole image into segments. Figure 9, it is evident the direct relationship between the size of the segments and the number of operations required to complete the segmented DINEOF: the smaller the segments, the fewer operations are required. However, it has to be pointed out that sensitivity analysis is important to verify the performance. As previously shown in Table 3, for the experimental settings analyzed in this article, the best segment size found during experimentation was 625 × 374 × 12. This analysis is suggested to be conducted on distinct scenarios, according to the particular requirements of the context.

Figure 9 Number of operations expected as the segments are reduced, or equivalently, the number of segments are augmented.

Conclusions

This research article introduced a new efficient approach to filling in missing high-resolution data from different satellite sensors. The proposed approach consists of three general steps: (1) Automatic satellite data download, (2) satellite data fusion, and (3) filling of missing satellite data using a computationally intensive approach. The first novelty of the proposed approach is that data from different sensors are used when performing satellite data fusion. As a second contribution, an approach is introduced to be able to process high-resolution images of large study areas. The proposed approach divides the orthomosaic into segments that DINEOF can process; the segments can be processed in parallel using a computer cluster; next, the orthomosaic is reconstructed without missing data. Finally, an analysis of the computational effort of the proposed approach was performed and it was found to be of quadratic order.

As proof of concept, the proposed approach was applied to fill in the holes in satellite imagery, using data from MODIS sensors aboard the Terra and Aqua satellite platforms and VIIRS aboard the SNPP and JPSS-1 satellites. The multi-sensor fusion approach implements geospatial techniques such as sliding averages and inverse weights interpolation with the squared distance and approaches to adjust the differences in the time each platform overflights the zone of study. Results showed that the proposed approach overcomes the traditional averaging strategy. When compared to the proposed approach, the traditional average strategy produces zones in which the values of Chl-a are overestimated. In contrast, the proposed approach preserves the oceanic structures required to study the ocean dynamics related to the currents and winds.

Although the DINEOF method is widely employed in several remote sensing studies to fill in missing data, the direct process of high-resolution images with DINEOF results in a high computational cost in terms of memory and computing power. For that reason, the proposed approach divides the merged image, and each segment is processed separately using DINEOF. Finally, the processed segments are assembled without missing data. The process of adjusting the input parameters for DINEOF endorses its performance to fill in missing data through the time series analysis. The analysis of the results shows, in general terms, that the best outcomes are obtained with low values of alpha = 3 and numint = 0.1. The proposed approach was implemented using Python 3 and Matlab R2018b, which enables the automation of repetitive tasks, including automated download of satellite data from the OBPG-Ocean Color Data for levels L2 and L3 of sensors MODIS and VIIRS. The characteristics of Python support the link between diverse platforms previously developed for satellite imagery: multiplatform and multiparadigm.

As future work, the proposed approach may be applied to distinct scenarios that may provide evidence of the efficiency of the proposed data-intensive approach. Between the application areas, one of the most relevant may be satellite-guided fishing through phytoplankton monitoring. In fact, phytoplankton constitutes the basic nourishment for small fishes, crustaceans, and other sea life forms that are the base food for larger fishes and sea mammals. Although the data-intensive approach was evaluated on sea monitoring, land applications may also be benefited from its efficiency. Additionally, with these results, it may be interesting to combine the proposed algorithm to address applications that incorporate artificial intelligence (i.e. forecasting, etc.). Finally, incorporating the algorithm in open libraries may favor the comparison with future proposals in this research area.

We thank the GHER group at the University of Liège for providing DINEOF; NASA OBPG for providing the satellite data used in experiments.

Additional Information and Declarations

Competing Interests

Author Contributions

Data Availability

The authors declare that they have no competing interests.

José Roberto Lomelí-Huerta performed the experiments, performed the computation work, prepared figures and/or tables, authored or reviewed drafts of the paper, and approved the final draft.

Juan Pablo Rivera-Caicedo conceived and designed the experiments, performed the experiments, performed the computation work, prepared figures and/or tables, authored or reviewed drafts of the paper, and approved the final draft.

Miguel De-la-Torre performed the experiments, analyzed the data, performed the computation work, prepared figures and/or tables, authored or reviewed drafts of the paper, and approved the final draft.

Brenda Acevedo-Juárez analyzed the data, prepared figures and/or tables, authored or reviewed drafts of the paper, and approved the final draft.

Jushiro Cepeda-Morales conceived and designed the experiments, analyzed the data, authored or reviewed drafts of the paper, and approved the final draft.

Himer Avila-George conceived and designed the experiments, performed the experiments, performed the computation work, prepared figures and/or tables, authored or reviewed drafts of the paper, and approved the final draft.

The following information was supplied regarding data availability:

The code is available at GitHub: https://github.com/jroberto37/fill_missing_data.git.

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
