# Peer review of "An approach to fill in missing data from satellite imagery using data-intensive computing and DINEOF"

_PeerJ Computer Science, doi:10.7717/peerj-cs.979_

## Round 0.1 · original submission · Major Revisions

Based on the reviewers’ comments and my own evaluation, I think the manuscript needs to be significantly improved before the consideration for publication. In particular, please clearly emphasize the novelty and justify better your results.

Reviewer 1 ·

Basic reporting

I support the publication of this work as the authors present a non-easy task through a creative approach. However, I suggest they follow the below suggestions before this research can be considered for publication.

I also noted that the quality of the figures provided is not the best and needs to be improved.

Experimental design

To improve the experimental design of filling missing (infrared/missing/clouds) satellite-data I suggested following a different approach. It relates to the use of non-missing data from model outputs and adding different percentages of clouds/missing data randomly. Better yet, this new approach can be developed on multiple time steps fields, thus you fully approach the strength of your method by following the existent nature of surface features.

Thus you will be able to develop stronger statistics (figure/table) to conclude how well/bad this new approach solves the filling of missing data while ensuring that your newly created data do not create false surface features and it is consistent with the original data.

It is also important that authors provide the original data and the methodology (Python/Matlab) scripts to recreate their analysis.

Validity of the findings

I believe the validity of the findings is questionable based on the current approach. See above for suggestions on a new approach to better evaluate statistics and skipping the creation of false features, which is conditional to some interpolation approaches.

Additional comments

I suggest you invest a little bit more time in this analysis and provide a robust analysis that we all in the satellite-data world will appreciate.

Reviewer 2 ·

Basic reporting

The article has a clear and unambiguous English. The English level is sufficient for an overall correct understanding of the text, taking into account that is not likely the mother tongue of any of the authors. However, I have highlighted (in my "General Comments") a few terms that should be replaced by more appropriate ones

Literature references are sufficient to illustrate the scope of the research and to give adequate background.

The structure of the article is correct and the authors share all the material needed for a thorough revision.

The objectives of the research and the results are clearly stated

Experimental design

The scope of the research fits in the aims and scope of the journal.

The authors propose a methodological improvement to well established gap-filling methods for ocean colour satellite data. The experimental design is correct and the obtained processor allows to apply the DINEOF program to high-resolution data through a segmentation procedure.

This is the main advance in the developed processor, together with a refinement in the merging procedure that, however, it is not clear to me how different is from the previous method, based on pixel averages (see my comment in the "General Comments" section).

The advance in knowledge is very small, but the results are of interest to the community of satellite data processing.

Validity of the findings

The results presented are insufficient to support the conclusions of the authors.
I think this is the weakest part of the article, that needs to be improved.
In my general comments below I suggest the authors different validation exercises that would serve to improve the soundness of the conclusions. In summary:

- For the assessment of the differences in the merging method, artificially removing a certain number of valid pixels in the original images is proposed. These pixels would be then used for validation

- Instead of the visual inspection in Figure 6, some statistical analysis is asked for

Additional comments

Below, my notes with comments and amendments requests throughout the text:

Abstract:
Line 19: Replace “capture” by “acquisition”
Line 23: Replace “For proof of concept” by “As a proof of concept”
Line 26: Replace “chlorophyll level” by “chlorophyll concentration”
Throughout the text, the term “acquisition” should be used preferably, instead of “capture”.

Introduction:
Overall, the introduction to the oceanographic background of the work needs a revision, preferably by an expert in the field, to correct several misunderstandings on the biological and physical basis of the phytoplankton role in Oceans.

Line 28: Replace “plant organisms” by “photosynthetic organisms”. Besides microscopic algae, Cyanobacteria are the main constituents of phytoplankton.
The references to the impact of phytoplankton are basically incorrect and should be removed or changed. The authors confound “weather” with “climate”. Moreover, the term “weather balance” is vague and meaningless.
Line 31: replace “biochemical cycle” by “biogeochemical cycle”
It is not clear how phytoplankton could provide information to predict impact of Climate change. The whole first paragraph needs to be revised.
Line 34: The most used abbreviation for Chlorophyll-a is “Chl-a”, instead of “Chlo-a”
Line 38: Replace “interaction” with “impact”. Chl-a interacts with light. Its interaction, in turn, determines the magnitude and shape of the reflectance spectra in water bodies.
Line 39: Replace “closed” by “close”
Line 42: replace “space” by “scene”
Line 53: What does “an overall temporal record” means?
Line 74: Again the confusion between “weather” and “climate”. The sentence in line 74 could be correct, because there are many meteorological satellites used in atmospheric modelling and weather forecasting. But the three references included are related to ocean color sensors and to the relationship of phytoplankton and climate

Methodology
Lines 89-90: The sentence has no sense: “importance of chlo-a in chlorophyll studies” ??
Line 103: What does “Python is a representative” means?
Line 110: Replace “passed by” by “passed over”
The formula for filling holes in the preprocessing step consists in calculating the average of the n nearest valid pixels (with data). Which is the difference in this calculation with the so-called “classical” method based on the average?
The adjustment applied to reduce the impact of differences in capture conditions is not clearly explained

Results
In the explanation of the merging procedure, it is stated that “the ortho-mosaic with most data related to chlorophyll is chosen and tagged as base-image”. However, in the example shown in the results, the VIIRS-SNPP image was taken as the base-image despite not being the one with more valid data. Can the authors explain the reasons for that choice?

Figure 5 shows that the classical method (GPT-SNAP) overestimates values with respect to the proposed method. But, why it is assumed that the proposed method is producing the correct values?
A good way of proving this is to produce artificial holes (by removing valid pixels with a simulated pattern) and then use the valid original pixels to test the performance of the merging procedure. Authors are encouraged to do this experiment.

Line 209: Replace “fussed” by “fused”
In the comparison described lines 209 to 215 (figure 6) it is very difficult to visually observe the differences in concentration claimed by the authors. A comparison of histograms and/or statistic test on differences would be more informative than the visual inspection.

Reviewer 3 ·

Basic reporting

The manuscript "An approach to fill in missing data from satellite imagery using data-intensive computing" by Rivera-Caicedo et al is well structured and the scientific investigation is clearly presented, including a basic context and motivation. However, the scoping is not clear and the methodology shows little originality.

The use of the DINEOF method for gap-filling is not new, and the literature contains other titles, including MODIS-based work.

Data Interpolating Empirical Orthogonal Functions (DINEOF): a tool for geophysical data analyses (2011)
Reconstruction of MODIS total suspended matter time series maps by DINEOF and validation with autonomous platform data (2011)
Analysis of gap-free chlorophyll-a data from MODIS in Arabian Sea, reconstructed using DINEOF (2018)
Exploratory Analysis of Urban Climate Using a Gap-Filled Landsat 8 Land Surface Temperature Data Set (2020)

Experimental design

Although the authors claim they introduce "a new approach for filling in missing data from satellite imagery", the level of originality is low. The DINEOF method is already well-known, and the authors present some references. For me it is not clear what is new, and the authors do not emphasize the novelty.

Validity of the findings

The validation of the findings is a very important part is such studies. I cannot find a distinct validation section, where the results are compared either with other methods or to measurements. This is a fundamental part which makes me have serios doubts about the quality of the work. The so-called new method is divided into modules, which in principle is good, but in fact only one module refers to data-gap filling, and the rest are basic image processing.

Additional comments

I strongly recommend to justify better your results, refer to more previous work and resubmit the manuscript.

Reviewer 4 ·

Basic reporting

The authors present an approach to fill in missing data from satellite
imagery using data-intensive computing. The approach was divided into three main modules. The idea is interesting; however, the authors must improve several details before accepting the paper.

The authors must improve the approach implementation explanation. There are gaps in how it was implemented. As explained, it seems that they only divided the data into smaller sets to be processed.

Also, several plots must be explained in another way. For example, in figure 5, the comparison is not understood.

the authors must justify parameters used in the implementation ( for example 3 x 3 window)

Table 1 is missing

Experimental design

The authors present interesting experiments and results. However, they must find a metric to validate the results; a visual comparison is not enough.

This metric will allow comparing the proposed approach with other methods.

Also, they can report processing time a compare it with the DINEOF algorithm processing time.

Validity of the findings

no comment

Additional comments

The idea is a novelty; however, it needs to be clarified to understand the real impact and originality.

---

## Round 0.2 · Minor Revisions

Compared with the previous version, the revised version has been obviously improved. However, as also pointed out by the reviewer, the authors are suggested to describe more details of the missing value imputation method to justify the main novelty.

So, in the related sections, such as the last paragraph in the section of Introduction, and the section of Method, authors are strongly advised to clearly point out the contributions and novelty when comparing with existing related work.

Reviewer 4 ·

Basic reporting

The authors present an approach to filling in missing data from satellite
imagery using data-intensive computing. The approach was divided into three main modules. The idea is interesting; however, the authors must improve several details before accepting the paper.

The authors must improve the approach implementation explanation a justify what is the main novelty of their method compared to the state of the art. As explained, it seems that they only divided the data into smaller sets to be processed and this is not enough.


Also, a computational complexity analysis must be added to understand and clarify the contribution of the approach.

Experimental design

Experimental design
The authors present interesting experiments and results. However, they must find a metric to validate the results; a visual comparison is not enough. This metric will allow comparing the proposed approach with other methods reported in the literature. for example the complexity of their approach.

Also, they can report processing time a compare it with the DINEOF algorithm processing time with previous work reported.

Validity of the findings

no comment

Additional comments

the authors must clarify the real impact and originality of the proposed method. Currently, this is not enough to accept the paper.

---

## Round 0.3 · accepted · Accept

Dear authors, after two rounds of revisions, the manuscript has been obviously improved. I think it is ready for the acceptance. Thank you so much for your careful revisions.